# OpenReview forum: "EReLELA: Exploration in Reinforcement Learning via Emergent Language Abstractions"
_ICLR.cc/2025/Conference — Submitted to ICLR 2025_

### Official Review · Reviewer_us7n · 2024-10-16

**Soundness:** 1
**Presentation:** 1
**Contribution:** 1
**Rating:** 1
**Confidence:** 5

**Summary:**

This paper investigates to what extent referential games, and the resulting emergent language abstractions, can be used to derive intrinsic rewards in hard exploration problems of reinforcement learning agents.

**Strengths:**

The research question of the paper is creative: can emergent language abstractions be used to help RL agents in hard exploration problems.

**Weaknesses:**

A) I don’t see empirical gains of using emergent language abstractions: Looking at Figure 2, it seems to me that the performance of the natural language abstraction agent (gray) and the emergent language abstraction agent (green) are within noise levels of each other. Thus, I find it difficult to conclude from the experiment that emergent language abstractions are important, in particular since both the natural language abstraction agent and the emergent language abstraction agent are using count based exploration terms as well. These should be ablated.

B) Even if there were empirical gains, I would want to see a comparison to state-of-the-art intrinsic reward methods for hard exploration problems based on natural language abstractions to see the point empirically proven that emergent language abstractions are supposedly preferable. In particular, I would expect comparisons to
- Mu et al. Improving Intrinsic Exploration with Language Abstractions. NeurIPS 2022. https://doi.org/10.48550/arXiv.2202.08938
- Klissarov et al. (2023). Motif: Intrinsic Motivation from Artificial Intelligence Feedback. arXiv. https://doi.org/10.48550/arXiv.2310.00166
- Zhang et al. OMNI: Open-endedness via Models of human Notions of Interestingness. ICLR 2024. https://arxiv.org/abs/2306.01711

C) Related to the above, I believe the authors need to evaluate on harder exploration problems, such as MiniGrid’s KeyCorridor-S3-R3 and MultiRoom-N10-S10, or MiniHack (Samvelyan et al. MiniHack the Planet: A Sandbox for Open-Ended Reinforcement Learning Research. NeurIPS 2021. https://doi.org/10.48550/arXiv.2109.13202). Moreover, I would like to see experiments beyond gridworlds, e.g., on Vizdoom (c.f. Henaff et al. Exploration via Elliptical Episodic Bonuses. NeurIPS 2022. https://doi.org/10.48550/arXiv.2210.05805).

D) I believe it would be important to add a tabula-rasa RL agent, as well as only RND baseline to Figure 2.

E) p8 Figure 3 looks to me like the experiments did not finish in time.

**Questions:**

- Abstract: It’s not entirely clear to me what limitations of NL-based counterparts you are referring to here.
- p1: “RND … which can be difficult to deploy” — Why are they difficult to deploy? RND is a very straightforward intrinsic reward method.
- From Figure 1 it looks like the intrinsic reward is only generated from the speaker. Why shouldn’t one also derive the intrinsic reward from the listener?

Comments
- p4 Figure 1 is too small to read. Same goes for other figures in the paper (e.g. Figure 2)
- p7 Figure 2 caption: explain the different methods variants in more detail.

What the authors would have to demonstrate to see an improved rating from me:
Demonstrate clearer gains of emergent language abstractions over natural language abstractions (A) on harder exploration problems (C) while also comparing to state of the art natural language abstraction methods (B) and adding tabula-rasa RL, as well as RND, baselines (D). Present results of finished experiments where each method is ran for the same number of steps (E).

---

> ### Author Response · Authors · 2024-11-28
> **Reply 1 (Part 1/2)**
>
> We thank the reviewer for their time and thorough review and constructive comments.
> We reply to those and the questions below.
>
> # Q1 : "Abstract: It’s not entirely clear to me what limitations of NL-based counterparts you are referring to here." :
> We assume you are referring to the following sentence: "Our results indicate that our proposed EL-guided agent, entitled EReLELA, achieves similar performance as its NL-based counterparts without its limitations."
>
> We meant to refer to the limitations of the NL-based RL exploration methods in terms of the cost in collecting NL descriptions.
> We agree with you that the formulation is ambiguous.
> We propose to rephrase by simply removing the ambiguous part of the sentence, since it is already addressed earlier in the abstract:
>
> "Our results indicate that our proposed EL-guided agent, entitled EReLELA, achieves similar performance as its NL-based counterparts."
>
> Please let us if you feel some precision would be better.
>
> # Q2: "p1: “RND … which can be difficult to deploy” — Why are they difficult to deploy? RND is a very straightforward intrinsic reward method."
>
> We agree with you, our statement about the complexity of deployment of RND and NGU here is unfortunately ambiguous and lacks precision.
> We meant it in comparison to a count-based exploration approach, mainly, in the sense that deploying a count-based exploration method can be considered simpler since it involves (i) fewer moving parts (.e.g state-count buffer versus e.g. RND's random and predictor networks, predictor optimizer ) that (ii) can also be deemed simpler to implement (no tricks required on the contrary to RND's tricks like reward normalization and observation clipping and normalization), and (iii) it involves fewer hyperparameters to finetune (e.g. only a reward-mixing coefficient as opposed to e.g. RND's reward mixing coefficient, architectures of random and predictor networks, hyperparameters of the predictor optimizer, different intrinsic and extrinsic discount factors, number of timesteps to step the initial random agent into the environment to harvest states for normalization purpose before starting optimization ...).
>
> In the revised paper, we have clarified this in the above specified terms.
> We hope that you agree with our statement now, but please let us know if you would prefer a different formulation or if you still find ways to clarify our statement.
>
> # Q3: "From Figure 1 it looks like the intrinsic reward is only generated from the speaker. Why shouldn’t one also derive the intrinsic reward from the listener?"
>
> We appreciate your attention to details in that matter. We have been considering deriving an intrinsic reward from the listener too, but the formulations we considered (e.g. entropy of the distribution over candidate stimuli, or referential game accuracy) forced our design to become an *across-training* exploration strategy [Stanton & Clune, 2018] (whereas the current design centered around the speaker's utterances allows an *intra-life* framing), primarily, and it also increased the complexity of the system further, secondly.
>
> We aim to investigate those other sources of intrinsic reward in subsequent papers. Indeed, we feel that the narrative of the current paper is already quite dense and we did not want risking losing the readers (more than we might already have in some places...) by adding an extra element to our proposed initial architecture.
>
> **With this paper, we emphasise that we solely seek to show viability in using EL to guide exploration of an RL agent, and therefore shows to reduce the designed architecture to its minimal requirements.**

---

> ### Author Response · Authors · 2024-11-28
> **Reply 1 (part 2/2)**
>
> # Comment (A)+(B): "Demonstrate clearer gains of emergent language abstractions over natural language abstractions [...] while also comparing to state of the art natural language abstraction methods"
> We thank you for listing clearly what we could address to raise your appreciation of the paper.
>
> Nevertheless, firstly, we disagree that it is necessary for the paper's results to demonstrate clearer gains of EReLELA compared to approaches using (synthetic) NL abstractions. Indeed, the paper does not claim superiority of EL-guided RL exploration methods over NL-guided ones, but rather solely that, in this minimal design, EL-guided RL exploration methods reach similar performance to NL-guided ones, and are therefore viable alternatives.
> Please let us know if that clarifies the issue, and whether you feel the paper should emphasise it further in any given part.
>
> Secondly, with regards to comparing to state-of-the-art NL-guided RL exploration methods, we agree that the comparison would strengthen our work, not towards showing superiority, but rather towards providing comparison grounds to put things into perspective. To that end, we are in the process of integrating and adapting to our codebase the codebases of [Mu et al., 2023] and [Raileanu et al., 2022]. We hope to include their results in our frameworks in the next revision of the paper.
>
> Please let us know if this answer fully addresses your concerns here.
>
> ## References:
> [Raileanu et al., 2022] : Raileanu, Roberta, and Tim Rocktäschel. "Ride: Rewarding impact-driven exploration for procedurally-generated environments." _arXiv preprint arXiv:2002.12292_ (2020).
> # Comment (C)+(D)+(E): "on harder exploration problems [...] and adding tabula-rasa RL, as well as RND, baselines  [...] Present results of finished experiments where each method is ran for the same number of steps.:
>
> We assume that the request to show results on harder exploration problems is especially relevant if the paper was to claim superiority of EL-guided RL exploration methods over state-of-the-art NL-guided ones.
> However, in light of the fact that our paper only seeks to show viability of EL-guided RL exploration methods, we hope that you can now appreciate why using the environments we have chosen, for we found them to strike a 'GPU-poor'-friendly balance between difficulty of the exploration problem and training time.
>
> That being said, experiments are currently running on *KeyCorridor-S3-R3* and *MultiRoom-N10-S10* as you proposed, along with tabula-rasa RL and RND baselines, but they are unfortunately not finished yet. We have not been able to include them in the current revision.
>
> Please let us know if this reply addresses your concerns on the matter.

---

### Official Review · Reviewer_2bD4 · 2024-10-27

**Soundness:** 2
**Presentation:** 2
**Contribution:** 3
**Rating:** 3
**Confidence:** 2

**Summary:**

The authors hypothesize that emergent languages can benefit RL agent exploration, to the same extent as expensive natural lanaguge descriptions. They propose a method to learn such emergent languages via reference games to induce intrinsic rewards jointly with the RL objective (EReLEA). They provide evidence that the learned emergent language is as useful, even more compact than natural language oracles.

**Strengths:**

1. Interesting method. The proposed CAM metric made a decent attempt at measurnig the quality of abstractions as far as I can understand its definition.
2. Contextualization of the problem is clearly articulated in sec 1 and 2.
3. The analysis about Zipf's Law of Abbreviation and the learned emergent langauge is insightful.
4. The ablation studies seem thorough (if only I can interpreate how they precisely differ in context).

**Weaknesses:**

My primary concern is presentation quality. Improved clarity can significantly benefit readability of this paper, as well as my understanding of the main method.
1. Typo/abbreviation mistakes: line 3 of the abstract "be done ne(?) with Reinforcement Learning", start of the first paragraph of introduction, line 42 "in effect" (?), single quotes in line 43, "it dynamics" -> "its dynamics" in line 98, and so on.
2. What's the superscript -1 on line 271?
3. Figure readability is sadly discounted by low resolution, small font size, and the lack of in-figure legends. Personally I find it hard to parse the results without clear legend names and matching color coding, even with captions.
4. Undefined H3.1 and H3.2 in lines 337-338
5. $\beta_1$ and $\beta_2$ in Figure 2 captions seem out of blue. Are they defined anywhere in the main text? Why do they imply "shared" and "anogostic"? Perhaps a table comparing configurations of parallel runs?
6. I appreciate the intuition, but I struggle to understand the formulation of the proposed CAM in sec 3.2. I think neither eq 4 nor the relative ambiguity of a language are CAM, but I cannot find exactly how CAM is computed in the main paper.

**Questions:**

1. What is the precise definition of CAM?
2. How do parallel experiments (i.e., different curves in Figure 2) differ exactly?

I am happy to raise the score if these questions are addressed with clarity.

---

> ### Author Response · Authors · 2024-11-28
> **Reply 1**
>
> We thank the reviewer for their time and thorough review and constructive comments.
> We have addressed all minor concerns in the revised paper and we reply to main comments and the questions below.
>
> # Q1: "What is the precise definition of CAM?"
>
> This concern was shared with Reviewer 4A62, we have provided a detailed reply to their review and invite you to centralise the discussion there.
> We provide here some specific details:
>
> We have added an algorithm to fully clarify the details of the CAM formulation, and reframed the Formalism paragraph with a more top-down narrative that we hope will be effective in enhancing the clarity but we are looking forward to more specific feedback if you have any further ideas about how to improve the matter.
>
> We have also added a CAM Distances paragraph at the end to clarify how we use the CAM measures in our analysis in Section 4.2.
>
> # Q2: "How do parallel experiments (i.e., different curves in Figure 2) differ exactly?"
>
> Thank you for your advice, as suggested, we have added a table summarising the different tested agent with their relevant parameters in Table 1.
>
> We are also further clarified what **shared** and **agnostic** mean at the end of Section 3.1.
>
> The parameters $\beta_1,\beta_2$ are not explained in the main text but solely in appendix G.1, as mentioned in the Agent paragraph of Section 4. We have added the mention towards explanations of the two hyperaprameters.
>
> We hope that these three additions fully answer your concerns on what the different settings are, but please let us know if you have any further advice to help us improve the clarity of the matter.
>
> # Comment 1: "Undefined H3.1 and H3.2 in lines 337-338"
>
> Thank you for catching this issue, we have now correct the matter and, as detailed to our answer to Reviewer 4A62, we have also improve the clarity of our Hypotheses paragraph by framing them with more precision to be testable.
>
> We hope that this clarifies the issue, but please let us know if you have any further advice.

---

> > ### Comment · Reviewer_2bD4 · 2024-11-30
> >
> > Hi Authors,
> >
> > Thank you for considering my suggestions and making edits that contribute to the paper's readability. Its current version is slightly more than the 10-page limit I think.
> >
> > I agree with R1(4A62)'s suggestion that less is more for presenting empirical results, and the same rule of thumb applies to formalism if I may add. For example, two pages on CAM seems too much. I would've placed only a clear algorithm box and one intuition paragraph. The authors sometimes elaborate too much on details and compromises ('despite', 'nevertheless', 'that being said') that prevent me from understanding the big picture. It is perfectly okay to move those behind-the-scene rationales or analyses to the appendix, which also helps highlight your key contributions.
> >
> > I remain conservative about the presentation quality (clarity of figures, preciseness of writing for the technical section). This limits me from confidently evaluating the soundness, understanding the results, and asking well-informed questions. There is no lower confidence level I can assign.
> >
> > In recognition of the authors' edits, the novelty (established before the technical section), and the well-known difficulty of getting emergent communication to work, I tend to raise the overall score from 3 to 4 if there were one. AC: Please note this and my low confidence score when considering my evaluation.

---

### Official Review · Reviewer_4A62 · 2024-11-06

**Soundness:** 2
**Presentation:** 2
**Contribution:** 2
**Rating:** 5
**Confidence:** 4

**Summary:**

This paper introduces an algorithm for augmenting reinforcement learning
algorithms with emergent communication-derived rewards that aid in tasks where
exploration is a difficult part of the task.  This algorithm works by training
agents to play a referential game with observations from the environment; the
speaker agent is then able to generate abstracted descriptions of the
observations for the RL agent which can encourage the agent to make new
observations that are not trivially different.  This algorithm is validated
with a handful of experiments and new metric "Compactness Ambiguity Metric" (CAM)
which quantifies the way in which the speaker agent generates abstract
descriptions of the environment observations.

**Strengths:**

The major strength of the paper is that the EReLELA algorithm itself is
presented clearly and is well motivated by (1) the success of language-based
abstraction methods for RL-based exploration and (2) the potential of emergent
communication to produce learned, human language-like communication.  I think
this contribution is especially important on the emergent communication side of
things because the field lacks practical applications of emergent languages,
and integrating it into an algorithm such as the one this paper introduces
could not only be effective in its own right but be an effective demonstration
of the applicability of emergent communication methods.

**Weaknesses:**

The major weaknesses of this paper are two fold.  First, CAM is not clearly
defined and/or justified.  It seems like it is a key analytical tool in the
empirical work of this paper, but its presentation did not give me a clear
picture of what it was doing either in theory or in practice.

The second also relates to clarity, namely the lack of clarity of the
experiments themselves.  The graphics themselves are quite noisy and refer to
settings that are not described in detail (e.g., "Agnostic STGS-LazImpa-10-1
ELA+AccThresh=90+Distr=256+UnifDSS").  Since the experimental settings are not
established at the beginning of the experiments section, I have very little
idea as to how to interpret the empirical results.  Is there a baseline?  Which
one is the proposed method?  Which other settings am I supposed to compare it
to?  Since I cannot easily answer these questions reading this section of the
paper, I cannot determine what is learned about the proposed algorithm.
I think it could be the case that the experiments themselves already contain
the requisite data for presenting an effective analysis of the algorithm, but
those things would need to be presented more simply and methodically.

**Questions:**

My main questions derive from the _Weaknesses_ section above: What are the
experiments showing?  Less is more when it comes to these graphics and
presenting these results.  Regarding CAM: what exactly is the metric?  What are
the inputs and outputs, precisely?  Once this is clarified, is it the case the
CAM is actually measuring the things we want to measure?  How do we validate
this?

It is possible I could be convinced to raise my review scores if the authors
are able to streamline the presentation of the experiments (especially the
graphics) _and_ the results are still substantive enough for the paper's
claims.  While I appreciate the thoroughness of the introductory sections,
I think they could be compact to make room for a more extensive explanation of
each experiment.  If the CAM and experiments sections of the paper had as much
clarity as the introduction, related work, and EReLELA sections, I easily
recommend acceptance.

### Minor Comments

- (Abstract) "done ne" -> "done"
- (1 Introduction) Typo at the very beginning?
- (1 Introduction) "NLs oracle" -> "NL oracle"
- (1 Introduction) In a sentence or two, why is it necessary to use
  language-based abstractions?  Wouldn't it be easier to represent things as,
  say, an embedding or more formal structure?  (I have an inclination as to
  what the answer to this question is, but I think it should be touched on in
  the text for clarity.)
- (Line 055) "NLs, that are" -> "NLs, which are"
- (Line 058) "hard-exploration" -> "hard exploration"
- (Line 065) What does "aligned by not similar to" mean?
- (Line 067) "advantages _over_ their NL"?
- (Line 090) The discussion of intrinsic versus extrinsic reward is a little
  unclear (partially on the writing level).  I can see what is being
  communicated, but someone with slightly less RL background might have a more
  difficult time.
- (Line 105) This is a good distinction to make.
- (Line 114) Extra space before ";"
- (Line 122) "entail to good exploration": Not sure what this means.
- (Line 138) "constraint" -> "constrain"
- (Line 160) Space after end quote
- (Line 161) Use `\citep`
- (Line 216) Extra space before ","
- (Line 228) Does "may not be passed" mean "is not passed with a certain
  probability"?  The phrasing "may not" is not clear here since it makes it
  sound like it is "not allowed to be passed".
- (Line 276) This paragraph is difficult for me to follow.
  - $i\in[0,N-1]$ suggests that $i$ is a real number, but I believe it is
    discrete.  Using $i\in{0, 1, \dots, N-1}$ would be clearer.
  - What is $\lambda_i$?
  - What is a "time interval threshold"?
  - Using pseudocode might be clearer here (I don't think I follow it enough to
    say this for sure, though).
  - (Sec 3.2) What is the input and output of CAM?  I get that it is creating
    a discrete distribution based on utterances used to describe observations,
    but what is the metric itself?  Is the distribution the metric itself or is
    it the entropy or the divergence from some baseline metric?
- While I appreciate explicitly naming the hypotheses, they are not stated with
  enough clarity and precision to be testable.  That is, how do we know
  precisely when the hypothesis as been validated or not?
- (Sec 4.1, Fig 2) These are difficult to follow, especially with the colors
  and the names which have not been well specified.  For example, I do not know
  what "shared" or "agnostic" refers to in the architecture.
  - Unless it is necessary, it would be good to reduce the number of
    referential game settings that you report so as to minimize confusion.
- The "natural language" baseline should be called a "synthetic language" since
  it is just programmatically generated and not gathered/derived from human
  language in a meaningful way.
- The different text colors for the experimental settings is a bit distracting.
  I think it would be better to come up with simple, easy-to-remember names for
  each setting and use those without worrying about colors (aside from the
  lines/legend on the plots themselves.

---

> ### Author Response · Authors · 2024-11-28
> **Reply 1 (Part 1/2)**
>
> We thank the reviewer for their time and thorough review and constructive comments.
> We have addressed all of your minor comments and more in the revised paper, thank you again for the level of details in that feedback.
> We reply to your main comments and the questions below.
> # Q1: "Regarding CAM: what exactly is the metric? What are the inputs and outputs, precisely? Once this is clarified, is it the case the CAM is actually measuring the things we want to measure? How do we validate this?"
>
> We have updated the CAM-related part in the revised paper, please let us know if it addresses your questions. For tractability, we reproduce below a summary of those changes:
>
>
> Firstly, we have enhanced the intuition-building paragraph in Section 3.1, in order to further clarifies motivations for the CAM. Please let us know if it can be further improved.
> Secondly, we have tried to clarify what are the $\lambda_i$ by detailing their motivations further in the following paragraph:
>
> "Next, we focus on the histogram that the metric returns. To sort compactness counts in this histogram, it is necessary to associate to each bin a partition of admissible compactness counts. Since compactness counts refer to time intervals, each bin of the histogram must refer to a range of time, between 0 and the maximum length $T$ of an RL trajectory/episode in the given environment. We assume that the start of the range associated with a given bin is the end of the range associate with the previous bin.
> Therefore, we can naïvely associate to each bin $i \in {0, 1,  \dots , N − 1}$ a time interval start $T_i$, defined relatively to the maximal length $T$. This framing is shown in Equation 3, with ⌈·⌉ being the ceiling operator. It is obtained by partitioning the whole range with the second and last hyperparameters $(\lambda_i)_{i\in\{0,1,\dots,N-1\}} \in [0,1]^N \; \text{ such that } \;  \forall(j,k), \; j<k \implies \lambda_j < \lambda_k$"
>
> ## What exactly is the metric? What are the inputs and outputs, precisely?
> The metric is highlighted in as many details as possible in the added Algorithm 1 of the revised paper. Please let us know if it is efficiently answering your questions. It takes as inputs the following:
>
> - $\mathcal{D}$: Dataset of $N_{\mathcal{D}}$ RL trajectories of length $T$;
> - $\text{Sp}_l$: Speaker agent for language $l$ being evaluated;
> - $N$: Number of histogram bins;
> - $(\lambda_i)_{i\in\{0,1,\dots, N-1\}} \in [0,1]^N$: partition hyperparameters;
>
> and it returns a histogram of compactness counts $H$.
>
> ##  is it the case the CAM is actually measuring the things we want to measure? How do we validate this?
>
> We performed interval validation of the CAM and presented it in Appendix E.1, but we appreciate that the original version of the paper did not emphasise it properly. We have now added the following lines to emphasise it:
>
> "In Appendix E.1, we show that this framing is sufficient to grant internal validity to our metric, meaning that this framing of the CAM (i) enables us to discriminate between different languages that are known to build different state-abstractions (e.g. synthetic languages that refers to all or only one specific attribute of objects, such as color or shape, used to caption a video stream that is egocentric viewpoint of an agent randomly walking in a 3D room with many randomly-placed objects), and (ii) maps languages without consistent state-abstractions (e.g. shuffled captions over a video stream) close to a null distribution histogram."
>
> Please let us know if those additions address your concerns and/or whether you have propositions to improve them.

---

> > ### Comment · Reviewer_4A62 · 2024-11-30
> >
> > I appreciate the edits that have been made, and I think they improve the paper.  In particular, I think the explanation of CAM is much better, and I now have a better idea of what that is.  I think the presentation of the experiments has also improved (e.g., Table 1), but the plots are still noisy and little hard to follow.  The main thing I would recommend on this front is to limit the varieties of EReLELA that are being experimented with---introduce as few varieties as are necessary to show the effectiveness of the result and perform the related ablation studies.
> >
> > I will move my overall assessment from 3 to 5, but no higher because the presentation of the experiments and their results does not leave me with a clear idea of what to take away (I would need to reread the full paper to be certain about this, but having skimmed the edits, I think this may still be an issue).
> >
> > Miscellaneous comment: CAM seems to be similar to Slow Feature Analysis (e.g., https://ieeexplore.ieee.org/document/9881217); referencing SFA could help contextualize things (only if it is actually related; if I am missing the point, then no need to reference SFA).

---

> ### Author Response · Authors · 2024-11-28
> **Reply 1 (Part 2/2)**
>
> # Q2:  "What are the experiments showing? Less is more when it comes to these graphics and presenting these results."
>
> We have enhanced our Hypotheses paragraph in order to try to clarify what the experiments must show and have taken greater care in their formulation towards being precision to be testable, please find it reproduced below for tractability:
>
> "We seek to validate the following hypotheses.
> Firstly, we consider whether a simple count-based approach over (synthetic) NL abstractions is sufficient to solve hard-exploration RL tasks **(H1)**.
> We refer to the corresponding agent using (synthetic) NL abstractions to compute intrinsic rewards as SNLA.
> We carry on with the hypothesis that a simple count-based approach over EL abstractions is similarly sufficient **(H2)**.
> In doing so, we will also investigate to what extent do ELs compare to SNL in terms of abstractions, using our proposed CAM.
> Using our proposed CAM, we consider two state abstractions to be aligned when their CAM distance is low.
> As the *MultiRoom-N7-S4* environment only shows differently-coloured doors in a partial observation context, the most important type of state abstraction is related to the colour of visible objects.
> On the other hand, since the *KeyCorridor-S3-R2* environment requires picking up an object behind a (unique) locked door, after having unlocked said door with a key, the most important type of state abstraction is related to the shape of visible objects.
> We consider a state abstraction to be meaningful in a given environment if it is aligned with the language oracle's abstraction that is the most important in said environment.
> Thus, we expect ELs to perform meaningful abstractions **(H3)**, i.e. being aligned with the colour-specific language's abstractions in the *MultiRoom-N7-S4* environment, and being aligned with the shape-specific language's abstractions in the *KeyCorridor-S3-R2* environment."
>
> We hope that those reformulations are sufficient towards streamlining the presentation of the results section, as we are not sure how we could further clarify our main 2 claims which are already addressed in 2 concise subsections, but we are looking forward to any specific advice if you have any more.
>
> We are also further clarifying what **shared** and **agnostic** mean at the end of Section 3.1.
>
> Please let us know if you find this sufficiently clear and precise, and/or whether you see ways to further enhance it all.
>
> ## Synthetic NL:
> We note also that we have clarified the issue around synthetic natural languages by adding the following discussion and renaming the originally 'natural language oracle' as 'synthetic natural language oracle' in order to clearly emphasise both the way those descriptions are obtained and the kind of grammar they rely on:
>
> "**Synthetic Natural Language Oracles.** Like Tam et al. (2022), we employ language oracles that provides NL descriptions/captions of the state. Like them, we mean to use the adjective ‘natural’ to specify the quality and form of the caption rather than the process in which it is obtained (i.e. programmatically as opposed to having human beings producing them). Nevertheless, in order to make the distinction clear, we will refer to those oracles as Synthetic Natural Language (SNL) oracles.
>
> That being said, we mean to emphasise that our considerations and results are agnostic to the process through which the NL captions are obtained, as we only indeed care about their quality and form, i.e. which vocabulary and grammar are being used, which here refers to that of the English natural language. We flag this as a limitation of our study because using NL captions produced from human beings would have yield a more varied and rich distribution, which would possibly impact the resulting RL agent’s performance (detrimentally supposedly). We make the choice here to only use synthetically-generated NL captions because they can be generated “accurately and reliably, and at scale” (Tam et al., 2022).
>
> Our implementation of SNL oracles are simply describing the visible objects in terms of their colour and shape attributes, from left to right on the agent’s perspective, whilst also taking into account object occlusions. For instance, around the end of the trajectory presented in Figure 6, the green key would be occluded by the blue cube, therefore the SNL oracle would provide the description ‘blue cube red cube’ alone. We also implement colour-specific and shape-specific language oracles, which consists of filtering out from the SNL oracle’s utterance the information that each of those language abstract away, i.e. removing any shape-related word in the case of the colour-specific language, and vice-versa."
>
> Please let us know if these changes are satisfactory, and/or whether you have any advice to help us further improve those matters.

---

### Meta-Review · Area_Chair_N4tD · 2024-12-19

**Metareview:**

The reviewers all agree this paper is not ready for publication. The authors should focus on improving the clarity of their writing. Most reviewers were confused by the discussion and explanation of their method and evaluation framework. Moreover, the authors should be clearer about what settings they foresee their method being of benefit compared to baselines (which their proposed method does not clearly outperform).

**Additional Comments On Reviewer Discussion:**

The reviewers' fair concerns around presentation quality, clarity of exposition, and comparison to similar previous methods were not sufficiently addressed by the authors' rebuttal.

---

### Decision · Program_Chairs · 2025-01-22

Reject